# S-farnesylation is essential for antiviral activity of the long ZAP isoform against RNA viruses with diverse replication strategies

**Dorota Kmiec, María José Lista, Mattia Ficarelli, Chad M. Swanson\*, Stuart JD Neil\***

Department of Infectious Diseases, King's College London, London, United Kingdom

\* chad.swanson@kcl.ac.uk (CMS); stuart.neil@kcl.ac.uk (SJDN)

**Data Availability Statement:** All relevant data are within the manuscript and its Supporting Information files.

## Abstract

The zinc finger antiviral protein (ZAP) is a broad inhibitor of virus replication. Its best-characterized function is to bind CpG dinucleotides present in viral RNAs and, through the recruitment of TRIM25, KHNYN and other cofactors, target them for degradation or prevent their translation. The long and short isoforms of ZAP (ZAP-L and ZAP-S) have different intracellular localization and it is unclear how this regulates their antiviral activity against viruses with different sites of replication. Using ZAP-sensitive and ZAP-insensitive human immunodeficiency virus type I (HIV-1), which transcribe the viral RNA in the nucleus and assemble virions at the plasma membrane, we show that the catalytically inactive poly-ADP-ribose polymerase (PARP) domain in ZAP-L is essential for CpG-specific viral restriction. Mutation of a crucial cysteine in the C-terminal CaaX box that mediates S-farnesylation and, to a lesser extent, the residues in place of the catalytic site triad within the PARP domain, disrupted the activity of ZAP-L. Addition of the CaaX box to ZAP-S partly restored antiviral activity, explaining why ZAP-S lacks antiviral activity for CpG-enriched HIV-1 despite conservation of the RNA-binding domain. Confocal microscopy confirmed the CaaX motif mediated localization of ZAP-L to vesicular structures and enhanced physical association with intracellular membranes. Importantly, the PARP domain and CaaX box together jointly modulate the interaction between ZAP-L and its cofactors TRIM25 and KHNYN, implying that its proper subcellular localisation is required to establish an antiviral complex. The essential contribution of the PARP domain and CaaX box to ZAP-L antiviral activity was further confirmed by inhibition of severe acute respiratory syndrome coronavirus 2 (SARS-CoV-2) replication, which replicates in double-membrane vesicles derived from the endoplasmic reticulum. Thus, compartmentalization of ZAP-L on intracellular membranes provides an essential effector function in ZAP-L-mediated antiviral activity against divergent viruses with different subcellular replication sites.

## Author summary

Cell-intrinsic antiviral factors, such as the zinc finger antiviral protein (ZAP), provide the first line of defence against viral pathogens. ZAP acts by selectively binding CpG

**Funding:** This work was funded by a Deutsche Forschungsgemeinschaft (German Research Foundation https://www.dfg.de) fellowship to DK (Project number: KM 5/1-1), Wellcome Trust Senior Research Fellowship (WT098049AIA) to SJDN (https://wellcome.org), and Medical Research Council (https://mrc.ukri.org) grant MR/S000844/1 to SJDN and CMS. This UK funded award is part of the EDCTP2 programme supported by the European Union. MF is supported by the UK Medical Research Council (MR/R50225X/1) and is a King's College London member of the MRC Doctoral Training Partnership in Biomedical Sciences. The funders had no role in study design, data collection and analysis, decision to publish, or preparation of the manuscript.

**Competing interests:** The authors have declared that no competing interests exist.

dinucleotides in viral RNAs, leading to their degradation or translation inhibition. Here, we show that the ability to target these foreign elements is not only dependent on ZAP's N-terminal RNA-binding domain, but additional determinants in the central and C-terminal regions also regulate this process. The PARP domain and its associated CaaX box, are crucial for ZAP's CpG-specific activity and required for optimal binding to the ZAP cofactors TRIM25 and KHNYN. Furthermore, a CaaX box, known to mediate post-translational modification by a hydrophobic S-farnesyl group, caused re-localization of ZAP from the cytoplasm and increased its association with intracellular membranes. The distribution of the long isoform of ZAP to intracellular membranes was essential for inhibition of both a ZAP-sensitized HIV-1 and SARS-CoV-2. Our work unveils how the determinants outside the RNA-binding domain assist ZAP's antiviral activity and highlights the role of S-farnesylation and membrane association in this process.

## Introduction

Cell-intrinsic antiviral factors are an important line of defence against viral pathogens. Although diverse in structure and function, these proteins often share common characteristics including broad antiviral activity conferred by targeting common aspects of viral replication, interferon-stimulated gene expression and rapid evolution due to selective pressures imposed by pathogens [1]. The zinc finger antiviral protein (ZAP, also known as PARP13 and encoded by ZC3HAV1) is a broadly active antiviral protein that is induced by both type I and II interferons and is under positive selection in primates [2–5]. It restricts RNA and DNA viruses as well as endogenous retroelements, with retroviruses and positive-strand RNA viruses being the most commonly used viral systems to study ZAP [6].

ZAP directly binds viral RNA to inhibit translation and/or target it for degradation [3,7–9]. ZAP interacts with several cofactors to restrict viral replication including the 3'-5' exosome complex, TRIM25, KHNYN and OAS3-RNase L [10–14]. There are four characterized ZAP isoforms, with the long (ZAP-L) and short (ZAP-S) isoforms being the most abundant [4,15]. All ZAP isoforms contain an N-terminal RNA-binding domain (RBD) and a central domain that binds poly(ADP)-ribose [3,7,16,17]. ZAP-L and ZAP-S differ in that ZAP-L contains a catalytically inactive C-terminal poly (ADP ribose) polymerase (PARP) domain [4]. ZAP distinguishes between self and non-self RNA at least in part by selectively binding to CpG dinucleotides [18–20]. These are present at a low frequency in vertebrate genomes due to events such as cytosine DNA methylation and spontaneous deamination of the 5-methylcytosine to thymine [21]. However, many vertebrate viruses, including RNA viruses that do not have a DNA intermediate, show CpG suppression in their genome and introducing CpG dinucleotides into the genomes of diverse viruses inhibits their replication [18,22–30]. While CpG dinucleotides in viral genomes may have multiple deleterious effects on replication [31–33], since ZAP has been shown to specifically bind CpG dinucleotides in viral RNA and restrict replication, it has been proposed that ZAP at least partially drives the CpG suppression observed in many vertebrate RNA viruses [18,34,35].

ZAP was originally identified as a restriction factor for murine leukemia virus and can target several different retroviruses including primary isolates of HIV-1 [3,35–38]. ZAP restricts HIV-1 replication when CpGs are enriched in the 5' region of HIV-1 *env* more efficiently than when CpGs are enriched in other regions of the viral genome and introducing CpGs into this region creates a highly ZAP-sensitive HIV-1 [18,33,35]. This model ZAP-sensitive virus has been used to characterize the role of CpG dinucleotides in ZAP-mediated restriction and

study its cofactors such as TRIM25 and KHNYN [13,18,19,34]. While the RNA binding domain (RBD) of ZAP is crucial for its selective antiviral activity [7,17,19,20], much less is known about the functional relevance of the other domains and motifs.

While there has been substantial progress in understanding how ZAP restricts viral replication, several key questions remain. First, the specific contribution of ZAP binding to CpGs for viral restriction is unclear, especially since ZAP can also restrict viruses with increased UpA abundance and CpG abundance does not always predict ZAP-sensitivity [14,15,30,39,40]. Second, while ZAP-L is often more antiviral than ZAP-S, depending on the experimental system and virus being studied, the specific determinants in the PARP domain that lead to increased antiviral activity and how this correlates with interactions between ZAP and cofactors remains to be resolved. Third, ZAP-L is targeted to the cytoplasmic endomembrane system and it is not clear whether this is required for inhibition of positive-strand RNA viruses that use these membranes as viral replication compartments or is a core aspect of ZAP antiviral activity against diverse viruses with different replication sites. To address these questions, we determined the functional relevance of the individual domains in ZAP and their contribution to antiviral activity against CpG-enriched HIV-1 and SARS-CoV-2. In addition to residues in the RBD that directly interact with a CpG dinucleotide, we identified that the PARP domain and CaaX box in ZAP-L are required for its antiviral activity against both viruses. Both the PARP domain and CaaX box were required for optimal interaction with ZAP cofactors KHNYN and TRIM25. Our findings explain the difference in activity between the two main isoforms of ZAP and highlight the functional contribution of C-terminal regions and proper subcellular localization for ZAP-L to control important human pathogens such as HIV-1 and SARS-CoV-2.

## Results

### Both the RNA-binding domain and C-terminal domains of ZAP contribute to its CpG specific activity

Full-length ZAP contains an RNA binding domain consisting of four CCCH zinc finger domains, a central domain comprised of a fifth CCCH zinc finger and two WWE modules plus a C-terminal PARP domain (Fig 1A) [3,4,16]. Early studies suggested that an N-terminal portion of the protein containing the four zinc fingers is sufficient for at least partial antiviral activity against diverse viruses or endogenous retroelements [3,8,36,41,42]. However, this was characterised using overexpression experiments in cells expressing endogenous ZAP and allowing for the endogenous and exogenous proteins to multimerise [17,43], complicating the experimental interpretation. To compare how much of ZAP's activity can be attributed to the RBD itself, we initially tested two truncation mutants of the protein, containing either the first 256 amino acids or the last 649 amino acids. Importantly, to eliminate potential interactions between endogenous and exogenous ZAP, these experiments were performed in ZAP CRISPR KO HEK293T cells [33]. Co-transfection of a full-length ZAP-L expression vector (pZAP-L) with wild-type HIV-1 (HIV-1$_{WT}$) resulted in a modest inhibition at the highest concentration. In contrast, a potent dose-dependent inhibition was observed for pZAP-L co-transfected with pHIV-1$_{env86-561}$CpG (a mutant containing additional 36 CpG dinucleotides introduced into *env* nucleotides 86–561 [13], referred to in this manuscript as HIV-1$_{CpG}$) (Fig 1B dashed lines). The N-terminal or C-terminal portions of ZAP-L were not sufficient to inhibit HIV-1$_{CpG}$ infectious virus yield, virion production or viral protein expression (Figs 1B and S1A).

Highlighting the importance of RNA binding for ZAP-L antiviral activity, deletion of the RBD or five alanine substitutions in the proposed RNA binding groove (V72/Y108/F144/H176/R189 – 5xRBM) [17,44] inhibited ZAP-L restriction of HIV-1$_{CpG}$ (Fig 1C). Mutating residues that directly interact with the CpG dinucleotide, such as Y108 and F144, has been

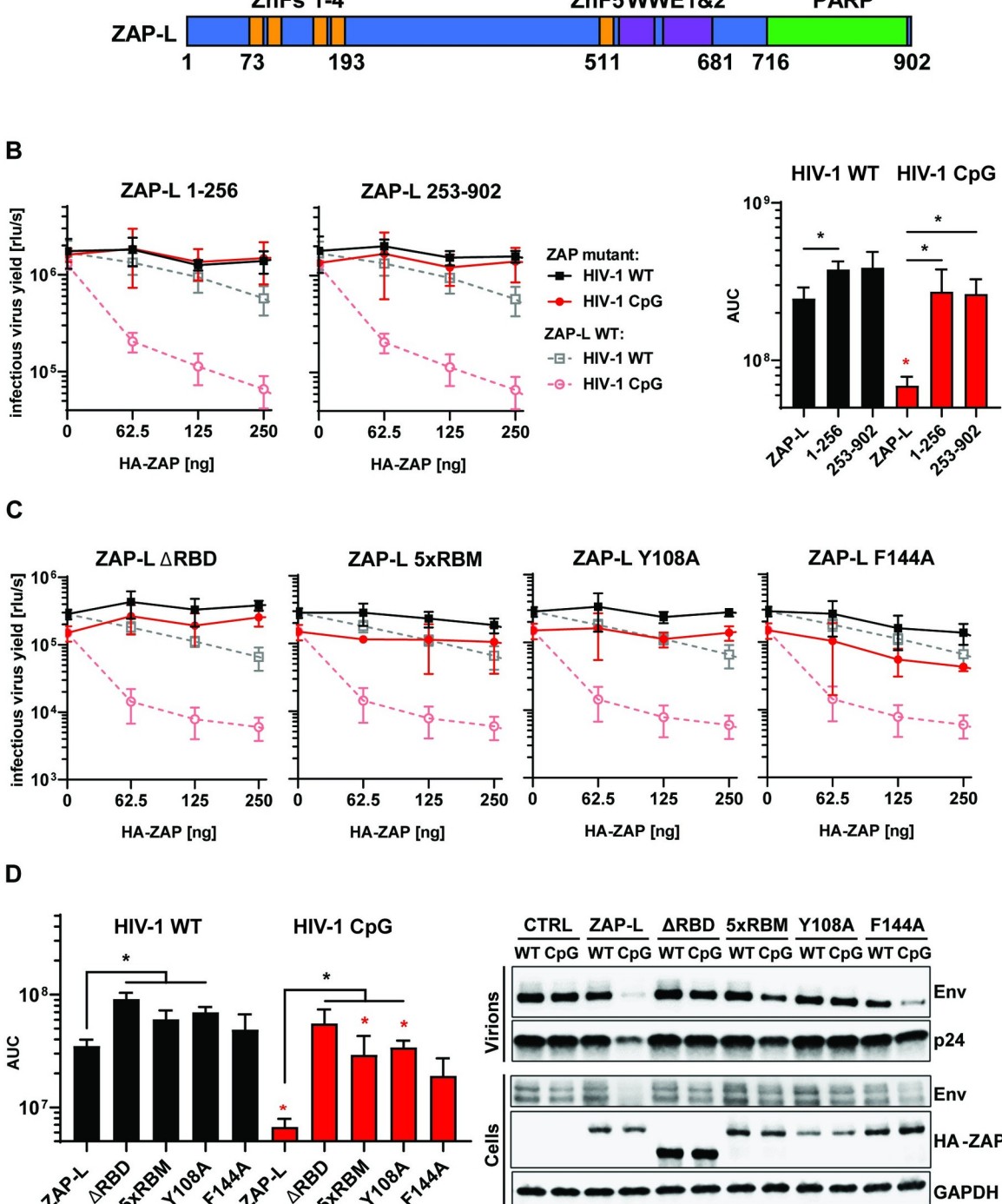

**Fig 1. RNA binding is crucial for ZAP's antiviral activity.** (A) Schematic showing domain organisation of long isoform of ZAP (ZAP-L): four N-terminal zinc fingers form RNA binding domain (RBD), fifth zinc finger (ZnF5) and two WWE domains are located in the central part and catalytically inactive Poly(ADP-ribose) polymerase (PARP) domain is at the C-terminal part. (B) Infectious virus yield measured by TZM-bl infectivity assay in relative light units per second [rlu/s] obtained from HEK293T ZAP KO cells co-transfected with wild type (WT; black) HIV-1 and CpG-enriched mutant (CpG-high; red) viruses and increasing doses of pcDNA HA-ZAP constructs encoding the full-length ZAP-L (dashed line), 1-256aa or 253-902aa parts of the protein (solid lines) (left panel). Area Under the Curve (AUC) calculated from the same titration experiments (right panel). (C) Infectious virus yield from HEK293T ZAP KO co-transfected with WT and mutant virus and increasing concentration of pcDNA HA-ZAP with truncated ZAP 253–902 (ΔRBD), ZAP-L mutant unable to bind RNA (V72A/Y108A/F144A/H176A/R189A; 5xRBM) or ZAP-L with substitutions at positions in direct contact with bound

RNA CpG (Y108A and F144A) and (D, left panel) derived AUC values. (right panel) representative western blot of produced virions and ZAP transfected (250ng) cells showing viral Env and Gag (p24) proteins as well as HA-tagged ZAP and GAPDH loading control. Mean of n = 3 +/- SD. * p<0.05 for HIV-1 CpG compared to HIV-1 WT for the same ZAP construct. * p < 0.05 for the comparisons demarked by the lines.

reported to partially reduce ZAP antiviral activity on CpG-enriched HIV-1 while restriction of wild type HIV-1 was increased, indicating that the CpG specificity for antiviral activity was eliminated [19]. In contrast, mutating Y108 abrogated ZAP antiviral activity for Sindbis virus [20]. Therefore, we tested the effect of ZAP Y108A on wild type or CpG-enriched HIV-1 and observed a large loss of activity on HIV-1$_{CpG}$ and no increase in antiviral activity for HIV-1$_{WT}$ (Fig 1C and 1D), which is consistent with the complete loss of function phenotype for this mutation observed for Sindbis virus [20]. ZAP-L F144A had moderate antiviral activity on HIV-1$_{CpG}$ and no activity on HIV-1$_{WT}$ (Fig 1C and 1D), indicating that it is a partial loss of function mutation. Thus, the ZAP RBD, and the specific residues that bind to the CpG, are necessary but not sufficient for restriction of CpG-enriched HIV-1, implying important effector functions elsewhere in the protein.

To determine domains required for ZAP antiviral activity outside the RBD, we tested ZAP mutants carrying deletions in the central domain (ZnF5 and WWE1 or WWE2) or the PARP domain (Fig 2A). While deletions in the central domain partly reduced ZAP antiviral activity, deletion of the PARP domain resulted in a large loss of activity against HIV-1$_{WT}$ or HIV-1$_{CpG}$ (Fig 2A and 2B). All human PARP proteins except for ZAP and PARP9 can catalyse the transfer of ADP-ribose to target proteins [45]. This lack of catalytic ability is due to a deviation from the conserved triad motif "HYE" required for NAD+ cofactor binding and PARP catalytic activity. Also the catalytic site present in active PARP domains is in a closed conformation in ZAP-L because the NAD+ binding pocket is occluded by a hydrogen bond between H810 and Y826 on one side and a short alpha helix between residues 803 and 807 at the other [46]. Three residues in the PARP-like domain have been found to be under strong positive selection in primates (Y793, S804, F805)—often a hallmark of pathogen-host interactions–and two of these are located in this alpha helix [4,46] (Fig 2C). The residues under positive selection are found on the surface of the domain and have been proposed to be a potential protein interaction site [46]. Interestingly, mutation of the residues that are present in the triad motif positions, Y787, Y819 and V876 to alanine or H-Y-E abolishes ZAP's inhibition of Sindbis virus [47], suggesting that the PARP-like domain provides important function.

To determine if these residues modulate antiviral activity against CpG-enriched HIV-1 and explain the apparent lack of inhibition by C-terminally truncated ZAP, we mutated ZAP's residues 787, 819, 876 (canonical triad positions, pink) and 793, 804 and 805 (sites under positive selection, green) within the PARP domain (Fig 2C). Mutation of the Y787, Y819 and V876 to alanine resulted in a large loss of antiviral function (Fig 2D and 2E) though this was also associated with a substantial decrease in ZAP expression (S1 Fig). Mutation of these residues to H-A-E did not alter ZAP expression but led to a significant loss of antiviral activity. Meanwhile, alanine substitutions at positions under positive selection did not affect the antiviral phenotype (Fig 2D and 2E). Therefore, the residues in the positions that constitute the triad motif in catalytically active PARPs, but not the rapidly evolving residues within the PARP-like domain, contribute to the restriction of CpG-enriched HIV-1. However, this does not fully account for the magnitude of the decrease in antiviral activity observed when the C-terminus of ZAP-L was deleted (ZAP-LΔPARP, Fig 2A and 2B).

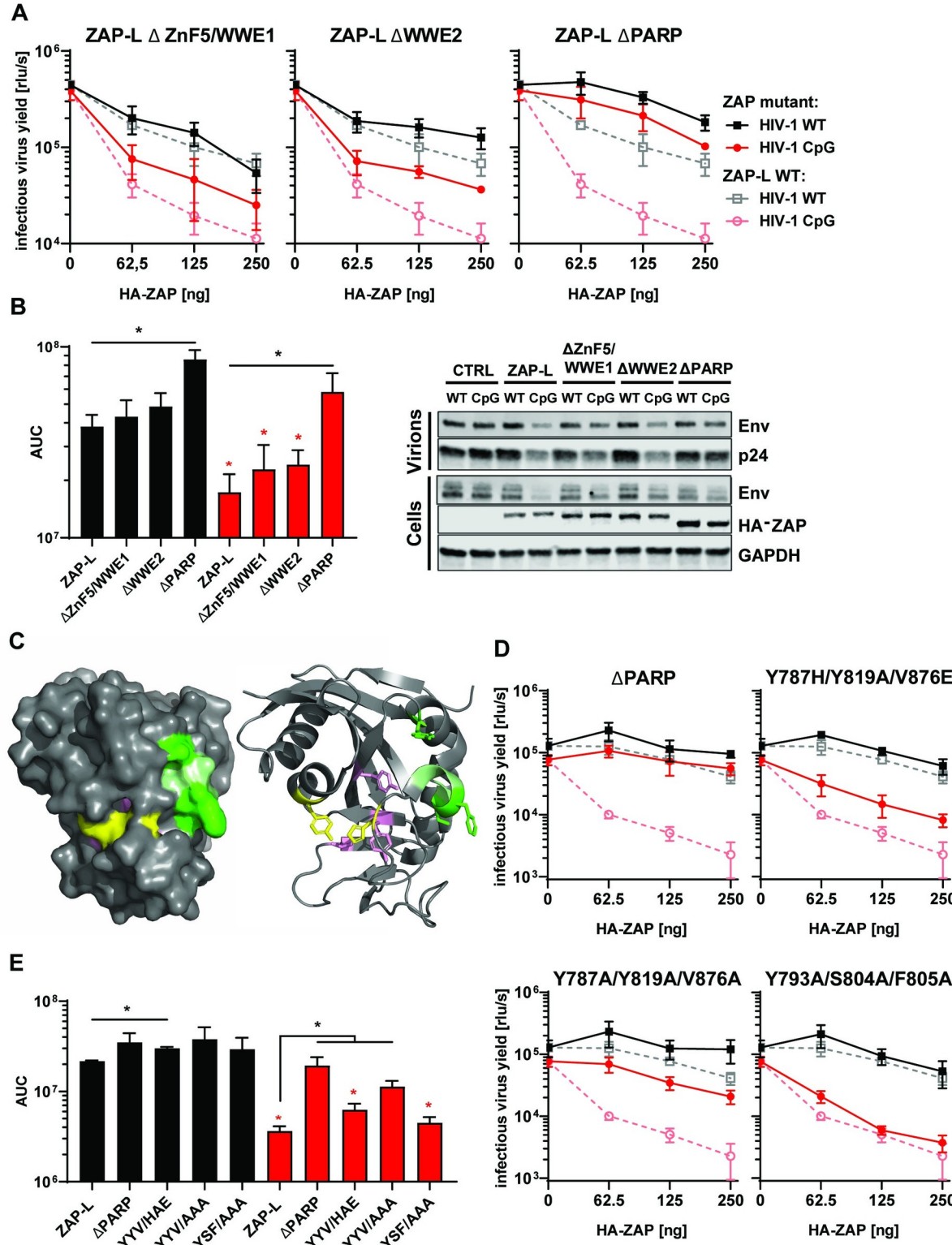

**Fig 2. Determinants of ZAP's function located outside RBD.** (A) Infectious virus yield from HEK293T ZAP KO co-transfected with WT (black) and mutant (red) virus and increasing concentration of pcDNA HA-ZAP-L control (dashed line) or mutated pcDNA HA-ZAP with deleted ZnF5 and first WWE domain (Δ511–563; ΔZnF5/WWE1), second WWE (Δ594–681; ΔWWE2) or PARP domain (Δ716–902; ΔPARP). (B) Corresponding AUC values and representative western blot (250ng). (C) Position of studied residues in crystal structure of ZAP's PARP domain. Residues under positive selection in primates are shown in green, canonical triad positions in pink and residues

forming the salt bridge which closes the NAD+ binding grove are shown in yellow. (D) Infectious virus yield from HEK293T ZAP KO co-transfected with WT (black) and mutant (red) virus and increasing concentration of pcDNA HA-ZAP-L control (dashed line), missing PARP domain or carrying amino acid substitutions in alternate triad motif (Y786H/Y818A/V875E, Y786A/Y818/V875A), or residues under positive selection (Y793A/S804A/F805A) (solid lines). (E) Corresponding AUC values. Mean of n = 3 +/- SD. * p<0.05 for HIV-1 CpG compared to HIV-1 WT for the same ZAP construct. * p < 0.05 for the comparisons demarked by the lines.

## The C-terminal CaaX box is crucial for ZAP antiviral activity against CpG-enriched HIV-1

While ZAP-L has been reported to be more active than ZAP-S, the relative activity remains unclear for many viruses and the experimental result could be affected by whether each isoform was expressed in the context of wild type cells expressing endogenous ZAP or in ZAP knockout cells [4,15,47–49]. Therefore, we compared the antiviral activity of both isoforms on HIV-1$_{WT}$ and HIV-1$_{CpG}$ in ZAP knockout cells. In agreement with the phenotype we observed for ZAP-LΔPARP (Fig 2), when ZAP-S was expressed in ZAP knockout cells it displayed no significant antiviral activity (Fig 3A and 3B). We also tested whether co-expression of both isoforms have additive or synergistic activity and found that ZAP-S had no additive effect against CpG-enriched HIV-1, even in the presence of ZAP-L (S2 Fig).

The ZAP-L PARP domain ends with a well-conserved CVIS sequence that forms a CaaX box (S3A Fig). This mediates a C-terminal post-translational modification through the addition of hydrophobic S-farnesyl group and ZAP-L S-farnesylation has been shown to be required for full antiviral activity on Sindbis virus [48,49]. This virus enters a cell through endocytosis and replicates its genome in cytopathic vesicles that are single membrane replication factories derived from the plasma membrane, endosomes, or lysosomes [50]. Therefore, it has been proposed that ZAP-L localizes to these membranes to target incoming viruses that enter cells through endocytic pathways or replicate in these membrane compartments [48,49]. However, HIV-1 does not replicate its genome or produce mRNAs in compartments formed from cellular membranes like classical positive-strand RNA viruses. Instead, its genome and mRNAs are produced from the integrated provirus using the same machinery as cellular mRNAs [51]. Furthermore, HIV-1 entry and the pre-integration steps can be bypassed by transfecting proviral plasmids, which will start the replication cycle at the gene expression step. To evaluate the contribution of S-farnesylation for HIV-1 restriction, we mutated the cysteine in the ZAP-L CaaX box to serine (C899S) and added the CaaX box to ZAP-S (ZAP-S + CVIS). The C899S mutation in ZAP-L completely abolished its antiviral activity on a transfected HIV-1$_{CpG}$ provirus while addition of the CaaX box to the C-terminus of ZAP-S resulted in a substantial increase in inhibition of HIV-1$_{CpG}$ (Fig 3B and 3C). Thus, the CaaX box is essential for ZAP antiviral activity on CpG-enriched HIV-1 and can substantially enhance ZAP-S activity even in the absence of the PARP domain. We also analysed whether the N-terminus of ZAP was sufficient for antiviral activity in the presence of the CaaX box. The CVIS motif was added to the first 256 or 352 amino acids of ZAP. The addition of the CVIS motif to ZAP 1–352 led to a partial increase in antiviral activity against HIV-1$_{CpG}$, comparable to that observed in the case of ZAP-S + CVIS (Fig 3D), though it did not add antiviral activity to ZAP 1–256, suggesting that there might be additional determinants of antiviral function present in the 256–352 region.

Because the closest paralogue to ZAP, PARP12, does not share the CpG-binding residues or CaaX motif found in ZAP (S3B and S3C Fig), we tested whether it inhibits HIV-1$_{WT}$ or HIV-1$_{CpG}$ and found that it had no antiviral activity on either virus (S3D and S3E Fig). Adding the CaaX box to PAR12 did not mediate antiviral activity against either virus (S3C, S3D and S3E Fig). However, a chimeric PARP12 protein containing the ZAP RBD in addition to the CaaX

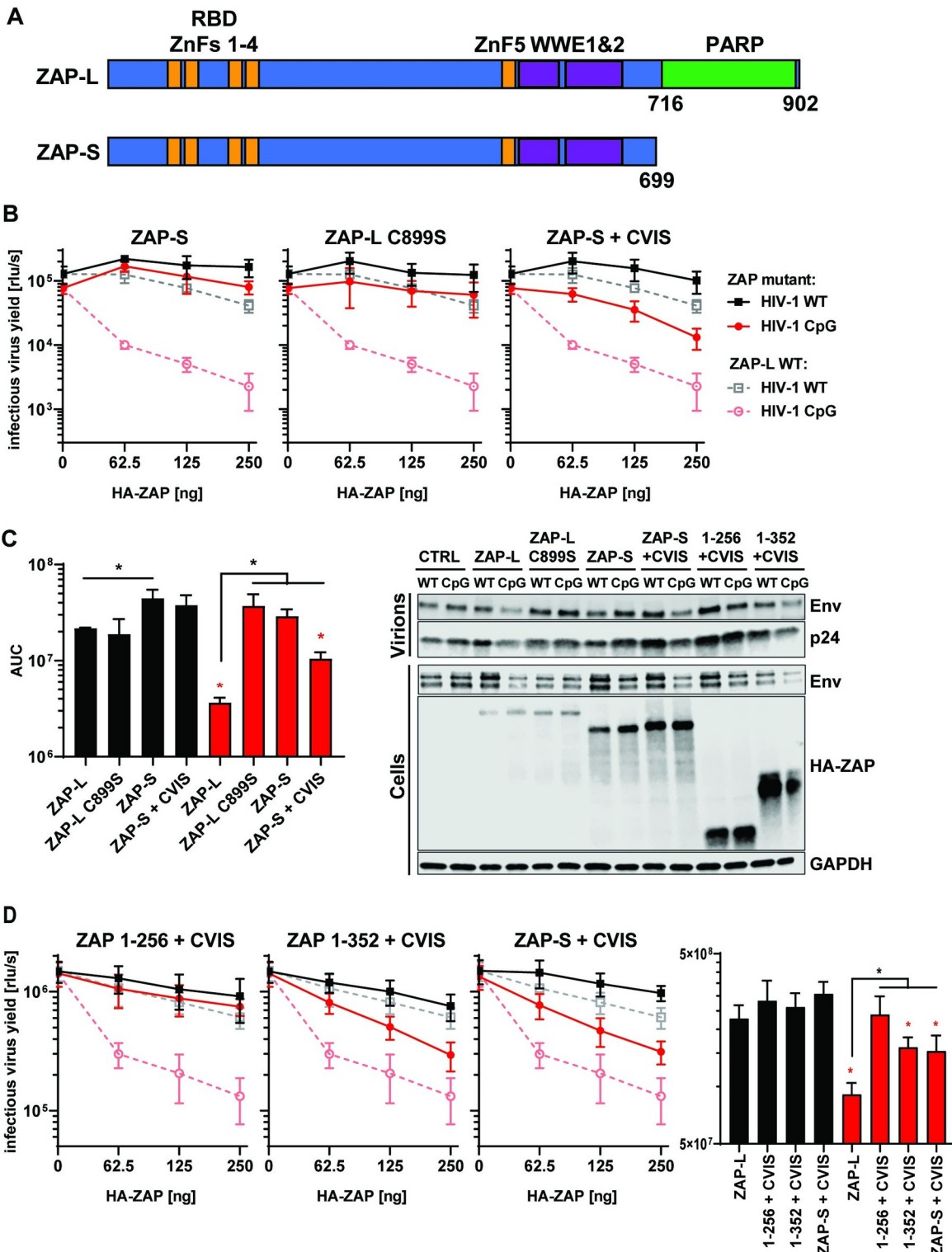

**Fig 3. Contribution of CaaX motif to ZAP's antiviral activity.** (A) Schematic showing ZAP-L, which contains PARP domain and CaaX motif (amino acids "CVIS") and short isoform of ZAP (ZAP-S). (B) Infectious virus yield from HEK293T ZAP KO co-transfected with WT (black) and mutant (red) virus and increasing concentration of ZAP-L (dashed line), ZAP-S, ZAP-L with mutated crucial cysteine (C899S) in CaaX or ZAP-S with added CVIS motif (solid lines). (C) Corresponding AUC values and representative western blot (250ng). (D) Infectious virus yield from HEK293T ZAP KO co-transfected with both viruses and pcDNA encoding truncated ZAP (1–256 or 1–352) with added CVIS motif and corresponding AUC values. Mean of n = 3–5 +/- SD. * p<0.05 for HIV-1 CpG compared to HIV-1 WT for the same ZAP construct. * p < 0.05 for the comparisons demarked by the lines.

box gained partial activity against HIV-1$_{CpG}$ (S3D and S3E Fig). This highlights that functional differences between these two paralogs in both the RNA binding domain and the C-terminal PARP-domain govern their antiviral activity.

As HIV-1 does not require intracellular membranes for producing genomic RNAs or mRNAs, the requirement for the CaaX box in ZAP-L for full antiviral activity is not likely to be due to it targeting specific sites of viral replication. However, the cytoplasmic endomembrane system could be used as a platform to establish an antiviral complex. To confirm that ZAP localization to membranes was dependent on the CaaX motif, we generated GFP-tagged versions of wild type and mutant ZAP. Importantly, the GFP-tag did not interfere with ZAP-L antiviral activity (S4 Fig). Confocal microscopy of live HEK293T ZAP KO cells transfected with GFP-ZAP (Fig 4A) showed that ZAP-S localized mainly to the cytoplasm, while ZAP-L accumulated in the intracellular vesicular compartments, confirming previous reports [48,49]. The localization pattern for ZAP-L and ZAP-S was reversed for ZAP-L C899S and ZAP-S + CVIS, respectively. Therefore, vesicular localization appears to correlate with antiviral activity for CpG-enriched HIV-1 (compare Figs 3B and 4A).

In addition to membrane-associated localization, ZAP-L has previously been shown to localize to stress granules, which are membraneless cytoplasmic structures, and localization to stress granules has been proposed to correlate with ZAP antiviral activity against Sindbis virus [44,52,53]. To determine if the CaaX box regulates ZAP-L localization to stress granules, we characterized the localization of the ZAP isoforms and mutant proteins that have differential subcellular steady state localization using poly(I:C) transfection to induce stress granules. As expected, GFP-ZAP-L and GFP-ZAP-S both localized to stress granules defined by G3BP puncta (S5 Fig). However, stress granule localization was not affected by mutation or transfer of the CaaX box, indicating that stress granule localization does not correlate with antiviral activity against CpG-enriched HIV-1.

To confirm that the ZAP-L membrane localization observed in the immunofluorescence experiments correlated with association with cellular membranes by subcellular fractionation, we isolated the cytoplasmic (C), membrane (M) and insoluble fractions (I) of HEK293T cells and found that ZAP-L, but not ZAP-S, was present in the membrane-enriched fraction (Fig 4B). This association could be disrupted by washing the cell lysates in 0.5M salt buffer (S6 Fig). Since this treatment did not affect membrane association of calnexin, ZAP S-farnesylation may mediate only a weak association with the cytoplasmic face of target membranes. Isolation of cytoplasmic and membrane fractions from ZAP knockout cells transfected with ZAP-L or ZAP-S expression vectors confirmed that while ZAP-L was present at comparable levels in both fractions, the ZAP-L C899S mutant protein had substantially increased cytoplasmic localization (Fig 4C). However, while ZAP-S-CVIS re-localizes to resemble ZAP-L localization (Fig 4A), its membrane association failed to survive the subcellular fractionation (Fig 4C), suggesting a weaker interaction. This, in keeping with its only partial gain of antiviral activity (Fig 3B and 3C), further indicates the importance of the integrity of the PARP domain in ZAP-L activity against CpG-enriched HIV-1.

We then determined whether ZAP targeting to intracellular membranes is required for it to interact with the cofactors required to mediate its antiviral activity against CpG-enriched HIV-1. Pulldown of GFP-tagged ZAP isoforms and mutants revealed that ZAP-L coimmunoprecipitated endogenous KHNYN more efficiently than ZAP-S, and the 1–256 and 1–352 truncation mutants bound even lower levels of KHNYN (Fig 5A and 5B). The same pattern was observed for TRIM25, which confirms previous observations that TRIM25 preferentially interacts with ZAP-L [11]. ZAP-S containing the CaaX box showed a gain of interaction with the cofactors. However, ZAP-L C899S bound more KHNYN than ZAP-S, indicating that both

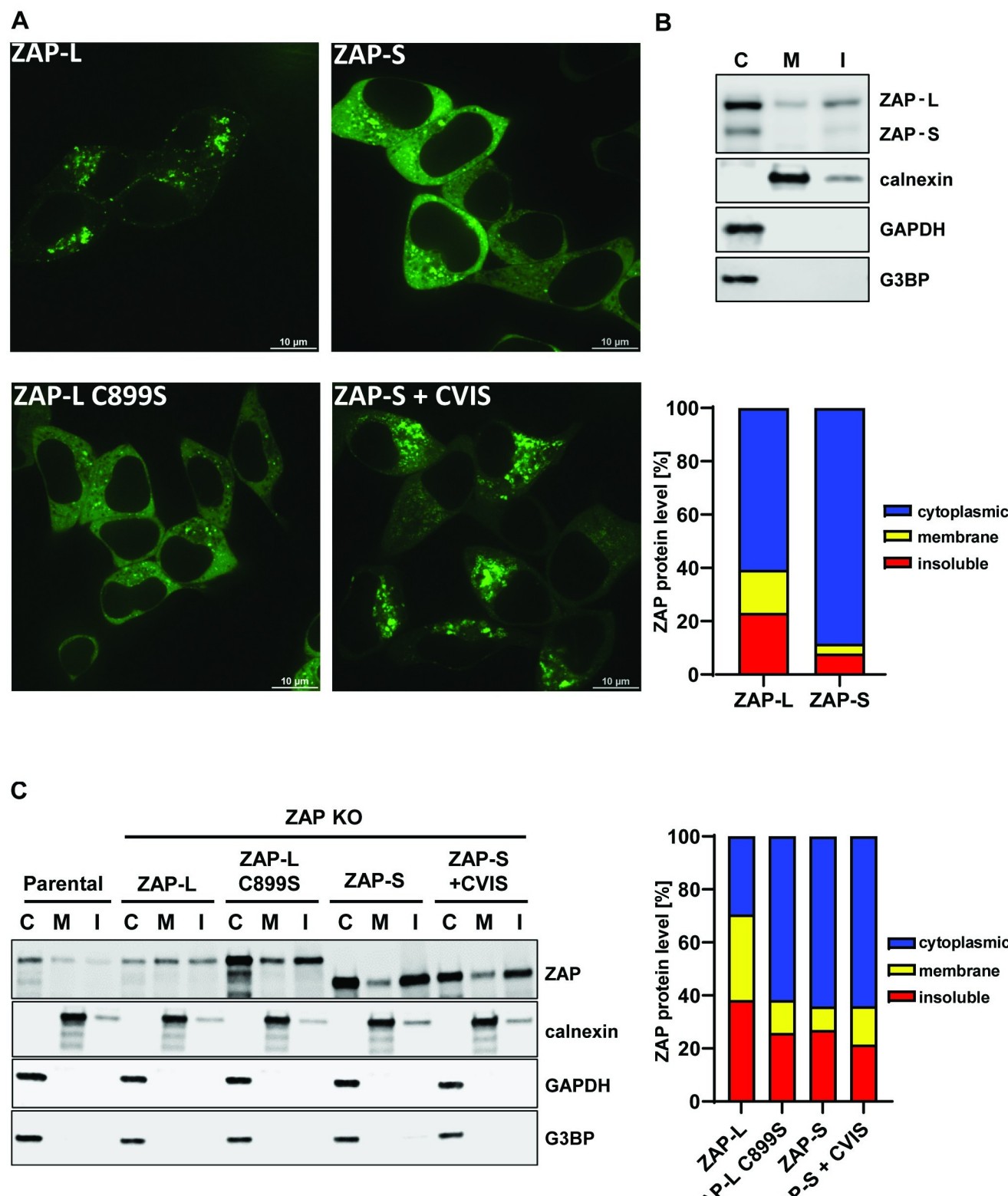

**Fig 4. Cellular distribution of ZAP-S, ZAP-L and their CaaX motif mutants.** (A) Confocal microscopy images of live HEK293T ZAP KO cells 24h after transfection with 250ng of GFP-tagged ZAP isoforms or ZAP-L with inactivated CaaX (C899S) and ZAP-S with added CaaX motif (+CVIS). Size bar 10µm. (B) Representative western blot and quantification of ZAP present in cell fractionation samples of parental HEK293Ts (mean of n = 5) or (C) ZAP KO cells

following transfection of 60ng HA-ZAP constructs (mean of n = 3). Cytoplasmic (C), membrane (M) and insoluble (I) fractions are shown. Calnexin serves as a marker for membrane protein and G3BP and GAPDH are cytoplasmic protein controls.

S-farnesylation as well as the PARP domain itself, likely play important roles in this interaction.

### The CaaX box is required for ZAP-L antiviral activity against SARS-CoV-2

Having established the determinants in ZAP required to restrict a virus that produces its RNAs in the nucleus, we sought to compare these data with SARS-CoV-2, which replicates exclusively in the cytoplasm. In contrast to the Sindbis virus replication compartments created from membrane invaginations in the plasma and endosomal membranes, SARS-CoV-2 replicates in double membrane vesicle compartments derived from the ER [54,55]. ZAP inhibits the replication of SARS-CoV-2 isolates representative of those found early in the pandemic, particularly after interferon γ (IFN-γ) treatment, which induces both ZAP-S and ZAP-L expression (Fig 6A) [5,56]. In late 2020, several SARS-CoV-2 variants of concern emerged, including B.1.1.7 (also known as alpha) [57,58]. This variant has been reported to be more resistant to type I interferon than the early lineage isolates [59,60]. Since ZAP is a component of the IFN-γ-mediated antiviral response [5], we tested whether this cytokine inhibited SARS-CoV-2 B.1.1.7. IFN-γ pre-treatment of A549-ACE2 cells inhibited an early lineage SARS-CoV-2 (England/02/2020) to a similar extent as the B.1.1.7 isolate 212, and both replicated better in ZAP knockout cells pre-treated with IFN-γ (Figs 6B and S7). This indicates that B.1.1.7 does not have decreased ZAP sensitivity. Of note, B.1.1.7 is defined by 14 non-synonymous mutations, 3 deletions and six synonymous mutations [58]. While CpG dinucleotides are strongly suppressed in SARS-CoV-2 [5,61], these large genomes still have >400 CpGs and the lineage-specific changes in B.1.1.7 lead to a decrease of only two CpG dinucleotides.

Both isoforms of ZAP have previously been reported to inhibit SARS-CoV-2 [5,56]. To test the ZAP determinants required to restrict SARS-CoV-2, we co-transfected ZAP knockout HEK293T cells with plasmids encoding human ACE2 and the indicated ZAP isoform or mutant protein, followed by infection with SARS-CoV-2 England/02/2020. Detection of intracellular N protein and viral RNA in the supernatants two days post-infection confirmed that both ZAP-L and ZAP-S restrict viral replication with similar efficiency, though ZAP-S is expressed at higher levels than ZAP-L (Fig 6C, 6D and 6E). Transferring the CaaX box to ZAP-S did not substantially increase its antiviral activity. Because the CpG abundance in SARS-CoV-2 is suppressed [5,61], it is not clear whether ZAP targets CpGs in the viral RNA or another motif. Since Y108 is a direct CpG contact residue that is required for ZAP antiviral activity on CpG-enriched HIV-1 (Fig 1C and 1D), we tested ZAP-L Y108A on SARS-CoV-2 and found a partial decrease in antiviral activity (Fig 6C and 6D). Similarly, mutating the YYV motif in place of the PARP catalytic triad moderately decreased ZAP-L activity against SARS-CoV-2. Importantly, ZAP-L restriction was abolished when the CaaX box was mutated, even though this mutant protein was expressed at higher levels (Fig 6C and 6D), highlighting the importance of S-farnesylation for ZAP-L to inhibit SARS-CoV-2. Thus, the determinants of restriction by ZAP-L are similar for a retrovirus that synthesizes its RNA in the nucleus and SARS-CoV-2, which replicates in compartments derived from the ER.

### Discussion

Herein, we have characterised the role of CpG binding residues in the RBD for ZAP-mediated restriction of HIV-1 and SARS-CoV-2 and how the C-terminal PARP domain and CaaX

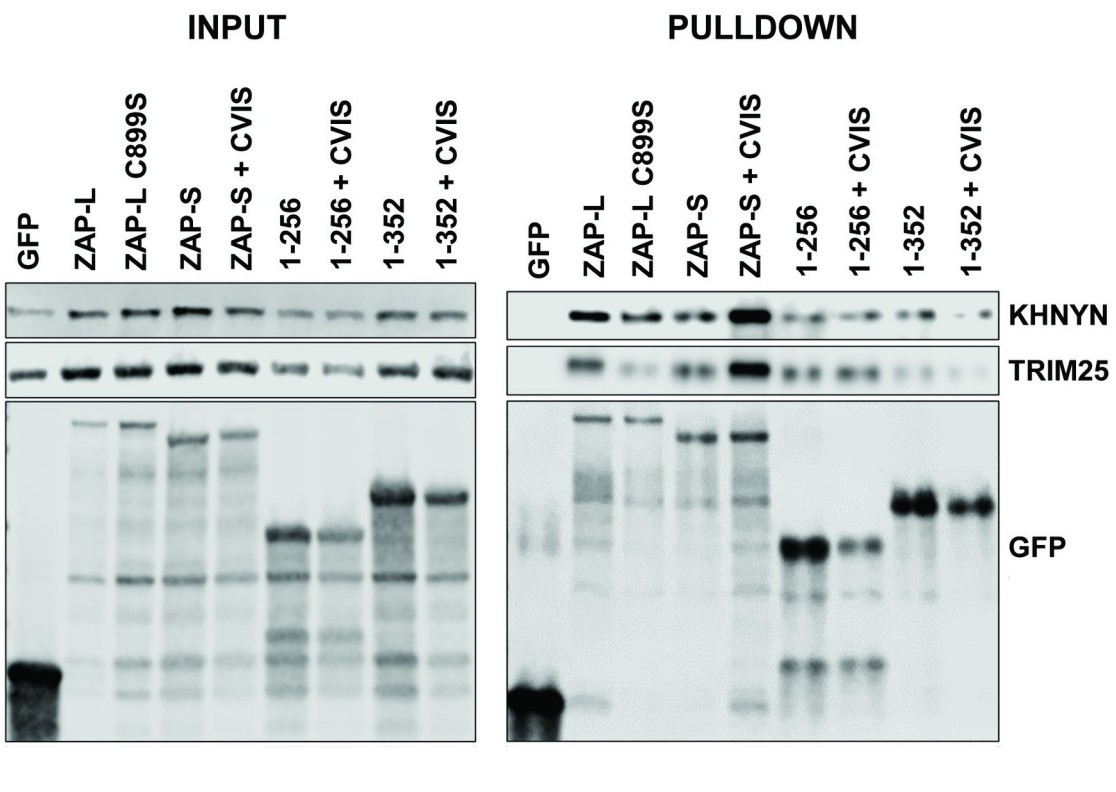

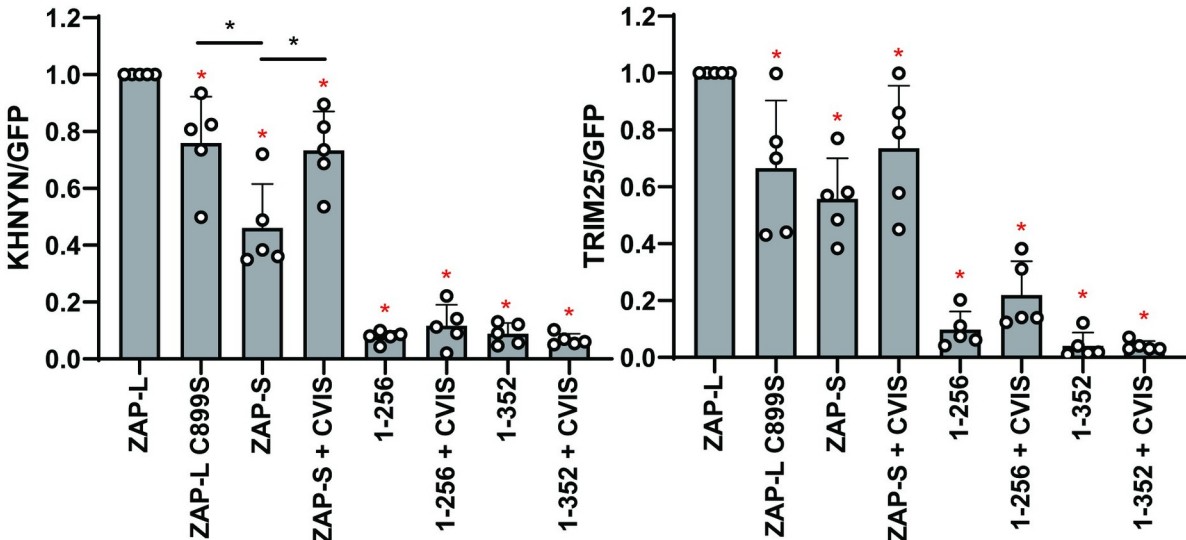

**Fig 5. ZAP cofactors KHNYN and TRIM25 bind most efficiently to wild type ZAP-L.** Upper left panel: Representative western blot of GFP-ZAP isoforms and mutant proteins expressed in HEK293T ZAP KO cells. Upper right panel: GFP-ZAP, TRIM25 and KHNYN co-immunoprecipitated using GFP-binding magnetic beads. Input and pulldown samples were stained for GFP as well as endogenous KHNYN and TRIM25. Lower panels: quantification of KHNYN and TRIM25 co-immunoprecipitated with GFP-ZAP normalized to the relative GFP signal. Mean of n = 5 + SD. * p < 0.05 for the sample compared to ZAP-L. * p < 0.05 for the comparisons demarked by the lines.

box promote the high antiviral activity of ZAP-L. The RBD is identical in both ZAP-L and ZAP-S. Within this four zinc finger domain structure, ZnF2 specifically accommodates a CpG in its binding pocket [19,20]. We found that mutating residues that directly contact the CpG

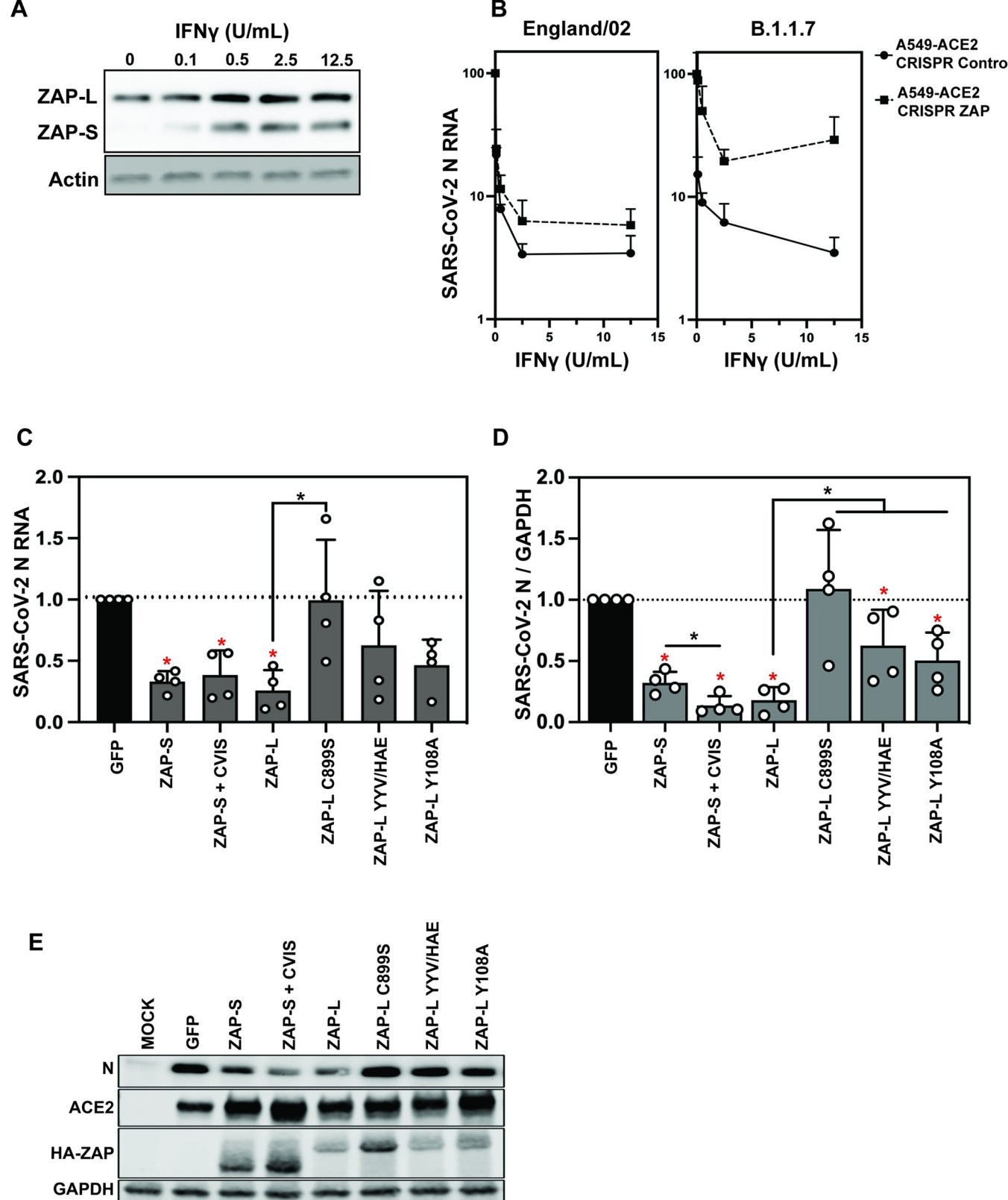

**Fig 6. ZAP-L requires the CaaX box to restrict SARS-CoV-2.** (A) Western blot showing IFNγ-mediated induction of ZAP-L and S in A549-ACE2 cells. (B) IFNγ treatment restricts early-pandemic SARS-CoV-2 strain England02 and B.1.1.7 variant of concern infection less potently in CRISPR ZAP cells than

CRISPR Control cells. SARS-CoV-2 RNA in the supernatant was measured by qRT-PCR. Mean of n = 3 + SD. (C) Viral RNA in the supernatant of HEK293T ZAP KO cells transfected with pcDNA encoding human ACE2 and indicated ZAP isoforms/ mutants or GFP control plasmid, 48h after infection with SARS-CoV-2 England 2 strain at 0.01 MOI. Quantification of qRT-PCR detecting viral nucleocapsid (N) RNA in the cell supernatant and (D) SARS-CoV-2 N protein levels in the infected cells, with a representative western blot (E). Mean of n = 4 + SD. * p < 0.05 for the sample compared to GFP. * p < 0.05 for the comparisons demarked by the lines.

decrease its antiviral activity against CpG-enriched HIV-1 and SARS-CoV-2. In contrast to a previous report [19], we did not observe a change in the specificity of ZAP for CpG-enriched HIV-1 compared to wild type HIV-1; we found that mutating Y108 and F144 decreased the overall antiviral activity of ZAP-L. The reason for this discrepancy is unclear, though the previous study ectopically expressed TRIM25 and ZAP constructs in a TRIM25/ZAP double knockout cell line. Mutating Y108 partially decreased ZAP-L antiviral activity for SARS-CoV-2, which could indicate that ZAP interacts with other motifs in the viral RNA in addition to CpGs. While it is clear that ZAP binds CpG dinucleotides in the context of single-stranded RNA, how the surrounding sequence and structural context of the CpG affect ZAP binding and antiviral activity as well as the role for non-CpG binding sites in viral RNA is still unclear [6,14,18–20,30,33,35,39,40].

Several studies have shown that ZAP-S has antiviral activity against diverse viruses including retroviruses, alphaviruses, Ebola virus, hepatitis B virus and SARS-CoV-2 [3,5,8,36,41,62]. However, many of these studies overexpressed ZAP-S in cells expressing endogenous ZAP-L and multimerization between the ectopic and endogenous ZAP could allow ZAP-S antiviral activity could mask the individual contribution of ZAP isoforms in viral restriction. By expressing ZAP constructs in ZAP knockout cells, we found that ZAP-S had little or no antiviral activity on HIV-1, though it was active against SARS-CoV-2. The antiviral activity of human ZAP-L and ZAP-S isoforms has been controversial [4,15,48,49]. While ZAP-L is usually more antiviral than ZAP-S, both have antiviral activity against some viruses when expressed on their own. For example, we show that ZAP-S appears to have little or no antiviral activity for CpG-enriched HIV-1 but it does inhibit SARS-CoV-2. Interestingly, some of the ZAP cofactors, such as TRIM25 and KHNYN, appear to interact better with ZAP-L than ZAP-S. However, ZAP-L and ZAP-S may form different antiviral complexes to target viruses with diverse replication strategies. In addition to TRIM25 and KHNYN, ZAP has been shown to interact with the 3'-5' RNA exosome complex, the deadenylase PARN, the decapping complex DCP1A-DCP2 and the 5'–3' exoribonuclease Xrn1 to target RNA for degradation [10,36]. Whether these proteins have preferential interaction with ZAP-L or ZAP-S is not known. In addition, both ZAP-L and ZAP-S also have been shown to regulate cellular mRNA expression and, depending on the transcript, the different isoforms could have differential activity [44,49,52,63], which could itself contribute to their differential activity against some viruses. ZAP-S has also been reported to regulate RIG-I signalling [64].

How determinants in the C-terminal domain of ZAP-L control its potent antiviral activity is crucial for a complete understanding of this antiviral pathway. The PARP domain is catalytically inactive but contains a C-terminal S-farnesylation motif, CVIS, that is required for full antiviral activity against HIV-1, SARS-CoV-2 and Sindbis virus [48,49]. This post-translational modification mediates ZAP-L localization to the cytoplasmic endomembrane system. While stress granules have been suggested as a site of ZAP's antiviral activity against Sindbis virus [53], the CVIS is not required for its location to these intracellular sites, implying that stress granules are not required for efficient ZAP-L restriction of CpG-enriched HIV-1 or SARS-CoV-2. Interestingly, the association between ZAP and cellular membranes appears relatively weak, in line with evidence that protein farnesylation itself is not sufficient for stable association with membranes [65,66]. There could be a dynamic exchange of ZAP-L between

membrane binding and the cytosol, which would allow ZAP-L to dynamically localize between membranes and stress granules [44,52]. ZAP-L has been shown to localize to endosomal compartments, the ER and nuclear membrane [44,48,49,67]. Appending the CaaX box to ZAP-S was sufficient to confer partial antiviral activity against CpG-enriched HIV-1, in agreement with previous Sindbis virus data [49]. As adding a CaaX box to ZAP-S does not confer full antiviral activity against CpG-enriched HIV-1, the catalytically inactive PARP domain appears to play an important role in ZAP-L function. This is consistent with the decrease in ZAP-L antiviral activity against CpG-enriched HIV-1 and SARS-CoV-2 when the residues in place of the catalytic triad in the PARP domain are mutated. Similar results have previously been reported for ZAP-L activity against Sindbis virus [47].

There are at least two potential hypotheses for why ZAP-L is localized to the cytoplasmic endomembrane system. First, it could localize to cytoplasmic membranes to target viruses that enter the cell through endocytosis or replicate in specific compartments derived from these membranes. This hypothesis has been proposed to explain why the OAS1 p46 isoform, which has a CaaX box and is prenylated, is more antiviral against several positive strand RNA viruses than the p42 isoform that does not contain a CaaX box [68,69]. This hypothesis is also consistent with previous studies demonstrating that ZAP-L is more active than ZAP-S against Sindbis virus [15,48,49]. However, a second hypothesis is that ZAP-L is targeted to cytoplasmic membranes to promote antiviral complex assembly with its cofactors, and our data is more consistent with this possibility. We have shown that the CaaX box is required for ZAP-L antiviral activity against CpG-enriched HIV-1 and SARS-CoV-2, which differ in their replication sites, and do not replicate in compartments formed from the plasma and endosomal membranes like Sindbis virus. In addition, ZAP-L targets RNA and DNA viruses with a broad range of replication sites and mechanisms [4,5,15,35,40,42,47–49,62,70–73]. Our data show that mutating the CaaX box moderately decreases ZAP co-immunoprecipitation with KHNYN and TRIM25, suggesting that membrane localization enhances the assembly of the ZAP-L antiviral complex. Of note, the PARP domain, and other regions outside of the RBD also appear to promote complex assembly. The nature of the ZAP antiviral complex is still poorly understood, with the specific cofactors required for the inhibition of different viruses and their localization yet to be elucidated. Defining the list of cellular proteins required for ZAP activity against a broad range of viruses with different replication strategies, solving the structure of the ZAP antiviral complex binding interfaces, and determining the specific subcellular sites of cofactor interaction with ZAP are necessary to elucidate how ZAP forms an antiviral complex to restrict viral replication.

## Materials and methods

### Expression constructs and cloning

Previously described pcDNA3.1 HA-ZAP-L and ZAP-S constructs [33] were rendered CRISPR-resistant by introducing synonymous mutations within exon 6. Primers were synthesized by Eurofins, and all PCRs were performed with Q5 High Fidelity DNA Polymerase (NEB). Monomeric enhanced GFP fused to N-terminus of ZAP via a flexible linker (GGGGSGGGGSGGGG) was synthesized by Genewiz and the full-length ZAP cDNA was reconstituted using an internal PsiI site. Specific mutations and deletions were generated using Q5 site-directed mutagenesis or Gibson Assembly (NEB) cloning. pcDNA3.1 HA-PARP12 was generated by PCR amplifying the PARP12 coding sequence (Dharmacon) and ligating into EcoRI/EcoRV sites of pcDNA3.1 using T4 DNA ligase (NEB). Construct sequence identity was confirmed by restriction enzyme digestion and Sanger sequencing (Genewiz). pHIV-

$1_{NL4-3}$ and pHIV-1$_{env86-561}$CpG in pGL4 were described previously [13,28]. pcDNA N-terminally C9-tagged human ACE2 construct was kindly provided by Dr Nigel Temperton.

## Cell lines and culture

Human Embryonic Kidney (HEK) 293T cells were obtained from the American Type Culture Collection (ATCC). Hela and HEK293T CRISPR ZAP KO (exon 6) cells were described previously [13,33]. TZM-bl reporter cells (kindly provided by Drs Kappes and Wu and Tranzyme Inc. through the NIH AIDS Reagent Program) express CD4, CCR5 and CXCR4 and contain the β-galactosidase genes under the control of the HIV-1 promoter [74,75]. A549-ACE2 were previously described [76]. Cells were cultured in Dulbecco's modified Eagle medium with GlutaMAX (Gibco) supplemented with 10% fetal calf serum (FCS), 100 U/ml penicillin and 100 µg/ml streptomycin, and grown at 37˚C in a humidified atmosphere with 5% $CO_2$.

## Transfection and HIV-1 infectivity assay

HEK293T ZAP KO cells (0.15–0.2mln) were seeded in 24-well plates and transfected the following day using PEI MAX (3:1 PEI to DNA ratio; Polysciences) with 500 ng pHIV-1 and 0–250 ng pcDNA3.1 protein expression construct. The total amount of DNA was normalized to 1 µg using pcDNA3.1 GFP vector. Media was changed the following day and cell-free virus-containing supernatants and cells were harvested two days post-transfection. To measure infectious virus yield, 10.000/well TZM-bl cells were seeded in a 96-well plate and infected in triplicate. Two days later, viral infectivity was determined using the Gal-Screen kit (Applied Biosystems) according to manufacturer's instructions. β-galactosidase activity was quantified as relative light units per second using a microplate luminometer.

## SARS-CoV-2 infection

hCoV-19/England/02/2020 (EPI_ISL_407073) or B.1.1.7 212 (hCoV-19/England/204690005/2020 (EPI_ISL_693401) were obtained from Public Health England. A549 cells constitutively expressing ACE2 expressing either a non-targeting sgRNA or ZAP targeting sgRNA that knocks out ZAP expression were seeded at 7.5 x $10^4$ cells/well in a 24-well plate. The cells were pretreated with IFN-γ for 16-hours and infected with each virus at MOI 0.01. Two days later the virus in the supernatant was harvested and quantified using qRT-PCR.

HEK293T ZAP KO cells (0.2 mln) were seeded in 12-well plates. The following day, the cells were transfected using PEI MAX with 100 ng pcDNA C9-ACE2 and either 400 ng pcDNA ZAP or GFP control vector. At 24 hours post-transfection, the cells were infected with hCoV-19/England/02/2020 at MOI 0.01 (prepared and tested as previously described in [76–78]. After 1 hour (h), cells were washed in PBS to remove the inoculum. Virus-containing cell-free supernatants and cell lysates were harvested two days later.

## Quantitative Real-Time PCR

RNA from infected cell supernatants was extracted using QIAamp viral RNA mini kit (Qiagen) and cDNA was synthesized using the High Capacity cDNA RT kit (ThermoFisher Scientific) following the manufacturer's instructions. The relative quantity of nucleocapsid (N) RNA was measured using a SARS-CoV-2 (2019-nCoV) CDC qPCR N1 probe (IDT DNA Technologies). qPCR reactions were performed in duplicates with Taqman Universal PCR mix (ThermoFisher Scientific) using the Applied Biosystems 7500 real-time PCR system. Relative SARS-CoV-2 RNA abundance was calculated by normalizing to a SARS-CoV-2 genome calibration curve using NATtrol SARS-Related Coronavirus 2 stock (ZeptoMetrix Corp).

## SDS-PAGE and immunoblotting

HIV-1 virions were concentrated by centrifugation at 18,000 RCF through a 20% sucrose cushion for 1.5 hours at 4˚C. Cells were lysed in radioimmunoprecipitation assay (RIPA) buffer containing cOmplete EDTA-free protease inhibitor (Roche) and 10U/ml benzonase nuclease (Santa Cruz). Cell lysates and concentrated virions were then reduced in Laemmli buffer and boiled for 10min at 95˚C. Samples were separated on gradient 8–16% Mini-Protean TGX precast gels (Bio-Rad) and transferred onto 0.45 μm pore nitrocellulose. Membranes were blocked in 5% milk and probed with mouse anti-HA (#901514, Biologend), rabbit anti-HA (#C29F4, Cell Signalling), rabbit anti-GAPDH (#AC027, Abclonal), mouse anti-G3BP (#611126, BD), rabbit anti-calnexin (#ab22595, abcam), rabbit anti-ZAP (#GTX120134, GeneTex), rabbit anti-GFP (#ab290, abcam), mouse anti-KHNYN (#sc-514168, SantaCruz), mouse anti-TRIM25 (#610570, BD), rabbit anti-SARS-CoV-2 N (#GTX135357, GeneTex), rabbit anti-ACE2 (#ab108209, abcam), mouse anti-HIV-1 p24 [79] or rabbit anti-HIV-1 Env (#ADP20421, CFAR), followed by secondary DyLight conjugated anti-mouse 800 (#5257S, Cell Signalling), anti-rabbit 680 (5366S, Cell Signalling), HRP conjugated anti-mouse (#7076S, Cell Signalling) or anti-rabbit (#7074S, Cell Signalling). HRP chemiluminescence was developed using ECL Prime Reagent (Amersham). Blots were visualized using LI-COR and Image-Quant LAS 4000 Imagers.

## Co-immunoprecipitation

HEK293T ZAP KO cells were seeded at 0.3–0.4 mln/ml in 10 cm dishes and transfected the following day with 10 μg pcDNA GFP or pcDNA GFP-ZAP plasmid using PEI MAX. Cells were harvested two days later and ZAP was immunoprecipitated using GFP-Trap magnetic agarose kit (Chromotek) following the manufacturer's instructions.

## Confocal microscopy

For live-cell microscopy, ~75.000 HEK293T ZAP KO cells were seeded onto poly-Lysine coated 24-well glass-bottom plates and transfected with 250 ng pcDNA3.1 GFP-ZAP using PEI MAX. Cells were visualized 24 h later using a 100x oil-immersion objective equipped Nikon Eclipse Ti-E inverted CSU-X1 spinning disk confocal microscope.

To visualize ZAP relocalization to stress-granules, ~50.000 Hela ZAP KO cells were seeded onto poly-Lysine coated 24-well glass-bottom plates and transfected with 125 ng pcDNA encoding GFP-ZAP using LT1 transfection reagent. 40 h post-transfection, cells were transfected with 100 ng poly(I:C) using Lipofectamine 2000 (Invitrogen) and fixed 6 h later in 2% PFA. Cells were blocked and permeabilized for 30min in PBS containing 0.1% TritonX and 5% Normal Donkey Serum (Abcam), stained overnight with mouse anti-G3BP (BD, #611126, 1:200 dilution), followed by 2 h staining with secondary donkey anti-mouse Alexa Fluor 546 antibody (Invitrogen, A10036, 1:500 dilution) and 1μg/ml DAPI.

## Cell fractionation

HEK293T and HEK293T ZAP KO cells (0.6–0.8 mln) were seeded in 6-well plates. The following day, ZAP KO cells were co-transfected using PEI MAX with 60 ng pcDNA HA-ZAP constructs and 940 ng pcDNA3.1 empty vector. Cells were harvested two days later, washed in PBS and processed using ProteoExtract Native Membrane Protein Extraction Kit (Sigma). Soluble cytoplasmic, membrane protein and insoluble fractions were isolated according to the manufacturer's instructions, with the addition of three 1 ml PBS or high salt washes between

extraction buffer I and II. The insoluble debris fraction was resuspended in RIPA buffer, sonicated and reduced in Laemmli buffer by boiling at 95°C for 10min.

### ZAP sequence analysis

Protein sequences of human ZAP-L orthologs were downloaded from the NCBI database (https://www.ncbi.nlm.nih.gov/). Sequences were aligned using ClustalW2 (https://www.ebi.ac.uk/Tools/msa/clustalw2/) and logo plots were generated using WebLogo online tool (https://weblogo.berkeley.edu/logo.cgi) (ref). ZAP PARP domain models were made using PyMOL based on PDB 2X5Y.

### Data analysis

The area under the curve (AUC) and statistical significance (unpaired two-tailed Student's t-test) were calculated using Prism Graph Pad. Data are represented as mean ± SD.

### Supporting information

**S1 Fig. Effect of expressed ZAP mutants on viral protein levels.** Representative western blots of the experiments shown in (A) Fig 1B and (B) Fig 2D.
(TIF)

**S2 Fig. Antiviral effect of ZAP-L and ZAP-S co-overexpression.** Infectious virus yield from HEK293T ZAP KO cells co-transfected with wild type (WT; black) HIV-1 and CpG-enriched mutant (CpG-high; red) viruses and increasing doses of pcDNA HA-ZAP constructs encoding ZAP-L (dashed lines), ZAP-S or 1:1 ratio of both isoforms up to 250ng each (solid lines). Values were normalized to infectivity in the absence of ZAP for each virus (100%). Mean of n = 5 +/- SD. Lower panel: representative western blot (250ng HA-ZAP).
(TIF)

**S3 Fig. Determinants of CpG-specific antiviral function in ZAP and PARP12.** (A) Logo plot of C-termini of mammalian and bird ZAP-L orthologues from NCBI database. Highly conserved serine 901 determines targeting by cellular farnesyl transferase which prenylates highly conserved cysteine 899. (B) Alignment of RNA-binding domains of human ZAP and its paralogue PARP12. Four zinc fingers (grey boxes) and ZAP residues interacting with CpG dinucleotide in bound RNA (highlighted in pink) are indicated. (C) Schematic showing the domain organisation of ZAP-L, PARP12 and PARP12/ZAP chimeric constructs. (D) Infectious virus yield from HEK293T ZAP KO co-transfected with WT (black) and mutant (red) virus and increasing concentration of pcDNA HA-ZAP-L CTRL (dashed line), PARP12, or ZAP/PARP12 chimera (solid lines). (E) corresponding AUC values and representative western blot (250ng). Mean of n = 3+/- SD. * p<0.05 for HIV-1 CpG compared to HIV-1 WT for the same ZAP construct. * p < 0.05 for the comparisons demarked by the lines.
(TIF)

**S4 Fig. CpG-specific antiviral activity of HA and GFP tagged ZAP isoforms.** Infectious virus yield from HEK293T ZAP KO cells co-transfected with wild type (WT; black) HIV-1 and CpG-enriched mutant (CpG-high; red) viruses and increasing doses of pcDNA ZAP with N-terminal hemagglutinin tag (HA) or monomeric enhanced green fluorescent protein (GFP) tag. Mean of n = 3 +/- SD.
(TIF)

**S5 Fig. Re-localization of ZAP isoforms and their CVIS mutants to stress-granules.** HeLa ZAP KO cells were transfected with 125ng GFP-ZAP (green) and stained for stress-granule

marker G3BP (red) following treatment with 100ng of poly(I:C). DAPI staining shows cell nuclei (blue).
(TIF)

**S6 Fig. Effect of 0.15M-1M NaCl washes on ZAP's membrane localization.** Western blot and protein quantification following fractionation of HEK293T cells. Cytoplasmic (C), membrane (M) and insoluble (I) fractions are shown, with relative levels of endogenous ZAP-L and ZAP-S, as well as controls calnexin (membrane fraction control), and G3BP, GAPDH and TRIM25 (cytoplasmic fraction controls).
(TIF)

**S7 Fig. ZAP depletion in A549-ACE2 cells.** Western blot of CRISPR control and ZAP CRISPR A549-ACE2 cells demonstrating that ZAP has been knocked out in the ZAP CRISPR cells.
(TIF)

# Acknowledgments

We thank other members of the Neil and Swanson laboratories for helpful discussions as well as Dr Monica Agromayor and Prof. Juan Martin-Serrano and their group members for advice and assistance with confocal microscopy. We thank Nigel Temperton for generously providing reagents. The following reagents were obtained through the NIH AIDS Research and Reference Reagent Program, Division of AIDS, NIAID, NIH: TZM-bl from Dr John C Kappes, Dr Xiaoyun Wu and Tranzyme Inc; HIV-1 p24 Hybridoma (183-H12-5C) from Dr Bruce Chesebro and Dr Hardy Chen. The Antiserum to HIV-1 gp120 #20 (ARP421) was obtained from the NIBSC Centre for AIDS Reagents.

# Author Contributions

**Conceptualization:** Dorota Kmiec, Chad M. Swanson, Stuart JD Neil.

**Data curation:** Dorota Kmiec, Chad M. Swanson, Stuart JD Neil.

**Formal analysis:** Dorota Kmiec, Chad M. Swanson, Stuart JD Neil.

**Funding acquisition:** Dorota Kmiec, Chad M. Swanson, Stuart JD Neil.

**Investigation:** Dorota Kmiec, María José Lista, Mattia Ficarelli.

**Methodology:** Dorota Kmiec, María José Lista, Mattia Ficarelli, Chad M. Swanson, Stuart JD Neil.

**Project administration:** Chad M. Swanson, Stuart JD Neil.

**Resources:** Dorota Kmiec, María José Lista, Mattia Ficarelli, Chad M. Swanson, Stuart JD Neil.

**Supervision:** Dorota Kmiec, Chad M. Swanson, Stuart JD Neil.

**Validation:** Dorota Kmiec, Chad M. Swanson, Stuart JD Neil.

**Visualization:** Dorota Kmiec, Chad M. Swanson, Stuart JD Neil.

**Writing – original draft:** Dorota Kmiec, Chad M. Swanson, Stuart JD Neil.

**Writing – review & editing:** Dorota Kmiec, Mattia Ficarelli, Chad M. Swanson, Stuart JD Neil.

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
