## [Decision Letter · Decision Letter 0]

6 Jul 2021

Dear Dr. Neil,

Thank you very much for submitting your manuscript "The C-terminal PARP domain of the long ZAP isoform contributes essential effector functions for CpG-directed antiviral activity" for consideration at PLOS Pathogens. As with all papers reviewed by the journal, your manuscript was reviewed by members of the editorial board and by several independent reviewers. In light of the reviews (below this email), we would like to invite the resubmission of a significantly-revised version that takes into account the reviewers' comments.

The reviewers were generally enthusiastic, but did find the data presentation difficult to follow. In addition to addressing their mostly minor comments, two of the reviewers questioned the need for some of the less interpretable data and suggested revising the manuscript into a more concise and focused paper.

We cannot make any decision about publication until we have seen the revised manuscript and your response to the reviewers' comments. Your revised manuscript is also likely to be sent to reviewers for further evaluation.

Sincerely,

David T. Evans

Associate Editor

PLOS Pathogens

Benhur Lee

Section Editor

PLOS Pathogens

Kasturi Haldar

Editor-in-Chief

PLOS Pathogens

orcid.org/0000-0001-5065-158X

Michael Malim

Editor-in-Chief

PLOS Pathogens

orcid.org/0000-0002-7699-2064

The reviewers were generally enthusiastic, but did find the data presentation difficult to follow. In addition to addressing their mostly minor comments, two of the reviewers questioned the need for some of the less interpretable data and suggested revising the manuscript into a more concise and focused paper.

Reviewer's Responses to Questions

**Part I - Summary**

Reviewer #1: Kmiec and colleagues show that the catalytically inactive poly-ADP-ribose polymerase (PARP) domain of the long ZAP isoform (ZAP-L) is essential for CpG-specific restriction of HIV-1 and SARS-CoV-2. Their results clarify the determinants outside the CpG RNA-binding domain of ZAP that are important for antiviral activity and highlight the role of S-farnesylation and membrane association in ZAP-mediated antiviral function. Overall, the experiments were well designed and performed and the main conclusion is supported by the results. The findings are important for better understanding the mechanisms of ZAP-mediated viral restriction. However, there are some limitations should be addressed (see below comments in details).

Reviewer #2: Kmiec et al show that the CaaX signal for farnyslation of ZAP-L is important for its antiviral activity for CG-rich HIV-1 and SARS-CoV-2. Importantly, they are able to confer this increased antiviral activity on ZAP-S by appending this signal. The mutants also correlated with increased affinity to ZAP co-factors KHNYN and TRIM25 and cellular localization to internal membranes. This is an impressive amount of work and is convincing. My major comments are in regards to the manuscript organization and clarification of what is new versus what is confirmation.

Reviewer #3: This paper reports interesting findings concerning the antiviral activity of ZAP, which targets viral RNAs. The major conclusion is that the C-terminal PARP domain (an enzymatically inactive one) contributes significantly to the activity, especially on the major target studied here, CpG-rich HIV viral RNAs. In addition, mutations affecting farnesylation also impact activity, likely by preventing correct intracellular localization to vesicular membranes. Both PARP domain and farnesylation are needed for normal interaction with cofactors TRIM25 and KHNYN. These findings are interesting and the data are solid though the presentation is hard to parse. I found many of the interpretations of the data to be unclear – I didn’t know what to make of the data in many experiments. I think the paper needs to focus on the more clear-cut results, to state the conclusions more simply, and give the harder-to-interpret data less time or weight.

The literature on the importance of the PARP domain for ZAP activity seems very confused (as noted, line 170), and I think this paper will add to the confusion. ZAP-L seems generally to have more activity than ZAP-S, but the fold increase seems to vary widely in different papers, most likely because different targets are being assayed (and maybe because different cells are being used, with different levels of expression of endogenous ZAP, a good point). Residues in the PARP active site (the inactive site) seem essential sometimes and yet not in the case of ZAP-S. There would seem to be an opportunity here to sort this out, or at least voice an opinion.

Overall I think the language creates confusion as to when a ZAP construct is said to have “CpG-specific activity”. Do we mean it has no activity on a CpG-low target? Or that it has some basal activity on a CpG-low target and much higher activity on a CpG-high target? And when a construct is said to have “no CpG-specific activity”, presumably it means it has roughly equal activity on CpG-high and -low targets – but how much activity does it have on these two – is it high or low, compared with a construct that is “CpG-specific” acting on a CpG-high target?

Maybe we need to be talking about a measure of selectivity – like the ratios of activity on the two targets? Or maybe we need to establish some kind of scale, or a standard to compare to? Maybe pick a maximal setting, call it 100%, and then talk about % of that activity that is seen with a given construct and target? Whatever, the present description is very confusing. I think when describing the activity of any mutant, we should always be told its activity on a CpG-high and a CpG-low target. Period.

One example of what is unclear is epitomized by a sentence in the introduction: (lines 91 ff.): “we identified that the PARP domain and CaaX box found in ZAP-L, but not ZAP-S, are required for antiviral activity against CpG-enriched HIV-1 and SARS-CoV-2, explaining why ZAP-L is much more antiviral against these viruses than ZAP-S”. If these domains were “required” then ZAP-S would have no activity. But ZAP-L has “much more activity”, so ZAP-S apparently has some. Which is it? Presumably “required” is too strong. Or maybe the second half of the sentence should be deleted.

Maybe one solution is to just talk about activity on the chosen CpG-high HIV substrate and basically forget the wild-type CpG-low HIV, where activity is apparently low anyway. Or else do the CpG-high first through all the mutants, make all the important points, and then go back and consider the CpG-low separately, where not much can be said anyway.

**Part II – Major Issues: Key Experiments Required for Acceptance**

Reviewer #1: 1. The whole study is dependent on ZAP KO HEK293T cells and transient transfection assays to evaluate antiviral effects of ZAP variants compared to WT proteins. It is important to know the levels of overexpressed ZAP compared to endogenous proteins in physiologically relevant cell types. For instance, CD4+ T cells for HIV-1 infection and airway epithelial cells SARS-CoV-2 infection. The authors can either perform some key experiments using these cell types or at least compare ZAP levels in transfected HEK293Twith these cell types.

2. Only one HIV-1 strain (WT NL4-3), a CpG-high HIV-1 mutant derived from WT NL4-3, and one SARS-CoV-2 strain (England 2) were tested. It would be helpful to test additional viral strains or a mutant SARS-CoV-2 containing more CpG.

Reviewer #2: 1. It is not easy to tell what is new here and what is confirmation of other studies either with different viruses or in different cells. The authors make a point that some previous studies could have been flawed because they used over-expression in cells that also express endogenous ZAP. However, I could not tell from the manuscript whether or not that really made a difference. For nearly every figure, the authors need to be much more clear about what are new insights here versus what are validation using a better experimental design.

2. Related to comment #1, I think the authors could better present what is new here by cutting down the figures and focusing the manuscript better. For example, is Figure 1 needed? Could Figure 2 be pared down and the more important parts moved to Figure 3? Do you really need some of the supplemental figures that are barely mentioned at all in the main text? I think a more concise and focused paper will help the authors in the long run. (I am not insistent that the material needs to be cut, but please consider the suggestion).

3. Figure 6: The increased amount of pull-down of TRIM25 and KHNYN by ZAP-L seems over-interpreted. The amount of increased pull-down may be statistically significant, but does not really seem to explain the extent of increased activity of ZAP-L relative to ZAP-S, and the ZAP-S+ CVIS does not attain the amount of binding of ZAP-L. It seems to me that there is more going on here. Finally, please work on the figure legend to make it easier to the data presented in the figure.

Reviewer #3: I see no need for major new experiments.

**Part III – Minor Issues: Editorial and Data Presentation Modifications**

Reviewer #1: 1. Fig. 1B legend: HIV-1 in the ZAP mutant should be HIV-1 WT. AUC plots (Fig. 1B, 1D, 2B, 2E and others) can be labeled better with virus strains used. Fig. 1D right panel: WB data show F144A significantly reduce Env expression in cells and virions. This point should be discussed.

2. Line 164: It seems that the sentence should read: “However, this does NOT fully account for the loss of phenotype…”

3. Line 197, cite figure number in the sentence.

4. Line 223: insoluble fractions (should be I not D) as showed in Fig. 5.

5. Fig. 6: A and B panel labels are missing.

6. Fig. 7: It would be more convincing if the authors show dose-dependent inhibition of SARS-CoV-2 by ZAP-L as they did for HIV-1 infection assays. Moreover, it would be helpful to show other SARS-CoV-2 protein expression in addition to the N protein.

Reviewer #2: 4. Line 80: The role of CG-richness is less clear for other viruses (e.g. for the alphaviruses or for MLV) than it is for the CG-rich HIV-1 targets used here and by others. The authors should qualify this statement to clarify what has been tested versus what is hypothesis.

5. Line 86: TRIM25 was first found to be a co-factor for ZAP in the alphavirus system and then for MLV, not the CG-rich HIV-1 system (https://journals.plos.org/plospathogens/article?id=10.1371/journal.ppat.1006145 and https://pubmed.ncbi.nlm.nih.gov/28202764/)

Reviewer #3: Specific issues:

The data in Fig. 1B and C really look different from the way they are presented in the text. While the ZAP WT control always looks nice, all the mutants frankly look inactive on any substrate. There’s no change with increasing ZAP. I don’t see that Y108A has any activity on either target – no different than the ∆RBD.

Fig 1D has bars for “ZAP-L”, but where is that data – is that the same as ZAP WT CTRL in B and C?

ll. 116 ff: A very confusing sentence. “Co-transfection of full-length ZAP….” with virus gave a modest inhibition of both normal HIV-1 and CpG-high HIV-1, but “wild type ZAP” gave potent inhibition of CpG-high HIV-1. What’s the difference between full-length ZAP and wild type ZAP??

ll. 124 ff: The description here seems unnecessarily complex: “the phenotype of RBD deletion could be phenocopied by five alanine substitutions in the proposed RNA binding groove”. Couldn’t one just say something like “A mutant with five alanine substitutions in the proposed RNA binding groove also had no activity”?

ll. 132: Statements like “the RBD of ZAP is essential yet insufficient for CpG-mediated restriction of HIV-1” may be true but doesn’t say if the RBD is sufficient for some activity – for some kind of non-CpG-mediated activity. What exactly do we mean by CpG-mediated activity vs. non-CpG-mediated activity? Some kind of basal activity?

ll. 137: Confusing: “deletion of the PARP domain resulted in an almost complete loss of CpG-specific inhibition”. So do we take it that the deletion had no effect on the activity against a CpG-low target? If so, we need to hear that. Maybe we need to see the high/low ratio, or levels of both activities.

line 152: I guess we know these triad motif mutations (in other settings) block PARP activity, but why do we think they disrupt “the structural integrity of the ZAP PARP domain”? I would have thought they were chosen not to disrupt overall structure.

While it seemed like a good idea that ZAP-S might only be working in concert with endogenous ZAP-L, the experiment to directly test this (line 174) seemed to eliminate this.

The PARP12/ZAP chimeras seem confusing. Not sure what is concluded from them. Are they helpful?

Line 219: What’s the conclusion of the poly(I:C) experiment? Presumably this is inducing stress granules or other ISGs, but why are the effects on ZAP localization not requiring the CaaX box? I didn’t understand how these results are interpreted.

Line 283: It wasn’t until this point very late in the paper that I was able to figure out that in this experimental system “ZAP-S has no antiviral activity on its own”. Maybe this could have been said earlier, along with the fact that it has activity in other settings. That alone would seem to be an important and interesting point to make, justifying some speculation as to the differences between the various assays.

Line 348: I don’t understand the logic of the sentence “the evolution of CpG-specific antiviral activity enhanced by the PARP domain in ZAP led to, or was a consequence of, the loss of its own ADP ribosylation ability.” Needs more explanation.

PLOS authors have the option to publish the peer review history of their article (what does this mean?). If published, this will include your full peer review and any attached files.

Reviewer #1: No

Reviewer #2: **Yes: **Michael Emerman

Reviewer #3: No
---

## [Decision Letter · Decision Letter 1]

7 Oct 2021

Dear Dr. Neil,

We are pleased to inform you that your manuscript 'S-farnesylation is essential for antiviral activity of the long ZAP isoform against RNA viruses with diverse replication strategies' has been provisionally accepted for publication in PLOS Pathogens.

Best regards,

David T. Evans

Associate Editor

PLOS Pathogens

Benhur Lee

Section Editor

PLOS Pathogens

Kasturi Haldar

Editor-in-Chief

PLOS Pathogens

orcid.org/0000-0001-5065-158X

Michael Malim

Editor-in-Chief

PLOS Pathogens

orcid.org/0000-0002-7699-2064

Reviewer Comments (if any, and for reference):

Reviewer's Responses to Questions

**Part I - Summary**

Reviewer #3: As in previous review

**Part II – Major Issues: Key Experiments Required for Acceptance**

Reviewer #3: I am satisfied with the revisions.

**Part III – Minor Issues: Editorial and Data Presentation Modifications**

Reviewer #3: (No Response)

PLOS authors have the option to publish the peer review history of their article (what does this mean?). If published, this will include your full peer review and any attached files.

Reviewer #3: No

---

## [Editor Report · Acceptance letter]

21 Oct 2021

Dear Dr. Neil,

We are delighted to inform you that your manuscript, "S-farnesylation is essential for antiviral activity of the long ZAP isoform against RNA viruses with diverse replication strategies," has been formally accepted for publication in PLOS Pathogens.

Best regards,

Kasturi Haldar

Editor-in-Chief

PLOS Pathogens

orcid.org/0000-0001-5065-158X

Michael Malim

Editor-in-Chief

PLOS Pathogens

orcid.org/0000-0002-7699-2064